# DRQA: Dynamic Reasoning Quota Allocation for Controlling Overthinking in Reasoning Large Language Models

## Abstract

Reasoning large language models (RLLMs), such as OpenAI-O3 and DeepSeek-R1, have recently demonstrated remarkable capabilities by performing structured and multi-step reasoning. However, recent studies reveal that RLLMs often suffer from overthinking, i.e., producing unnecessarily lengthy reasoning chains even for simple questions, leading to excessive token consumption and computational inefficiency. Interestingly, we observe that when processing multiple questions in batch mode, RLLMs exhibit more resource-efficient behavior by dynamically compressing reasoning steps for easier problems, due to implicit resource competition. Inspired by this, we propose *Dynamic Reasoning Quota Allocation (DRQA)*, a novel method that transfers the benefits of resource competition from batch processing to single-question inference. Specifically, DRQA leverages batch-generated preference data and reinforcement learning to train the model to allocate reasoning resources adaptively. By encouraging the model to internalize a preference for responses that are both accurate and concise, DRQA enables it to generate concise answers for simple questions while retaining sufficient reasoning depth for more challenging ones. Extensive experiments on a wide range of mathematical and scientific reasoning benchmarks demonstrate that DRQA significantly reduces token usage while maintaining, and in many cases improving, answer accuracy. By effectively mitigating the overthinking problem, DRQA offers a promising direction for more efficient and scalable deployment of RLLMs, and we hope it inspires further exploration into fine-grained control of reasoning behaviors.

## 1 Introduction

Reasoning large language models (RLLMs), such as OpenAI-O3 (OpenAI, 2025) and DeepSeek-R1 (DeepSeek-AI et al., 2025), have recently showcased remarkable capabilities in complex problem solving and decision-making, achieving state-of-the-art performance across a wide range of tasks. However, recent studies have revealed that LLMs often generate unnecessarily lengthy reasoning chains, even for simple questions like "2+3=?" (Sui et al., 2025; Chen et al., 2025). While extended reasoning can improve accuracy on complex tasks, this tendency to *overthink* leads to excessive token usage and growing computational and economic costs, posing significant challenges for the scalable and practical deployment of RLLMs in real-world scenarios.

Inspired by recent findings on instruction-tuned LLMs (Lin et al., 2024; Cheng et al., 2023), which show that processing multiple inputs together during *batch inference* can reduce the total generated length compared to answering them individually, we investigate whether a similar phenomenon exists in RLLMs. Our study reveals that this effect in RLLMs goes beyond mere solution shortening: batch inference also *compresses the chain-of-thought reasoning process* itself. For example, as shown in Figure 1, answering three questions together yields only 648 tokens in total, compared to 1205 tokens when answered separately. This suggests that under a shared context window, questions implicitly compete for a global reasoning quota, prompting the model to prioritize essential logic and suppress redundancy, an effect we refer to as "*resource competition pressure*".

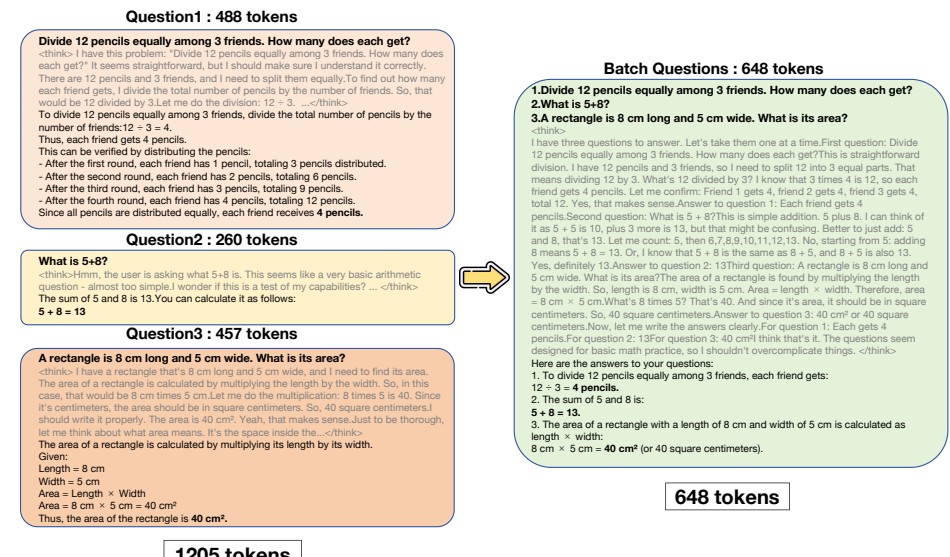

Figure 1: Comparison between batch inference and single-question inference using Deepseek-R1. Answering three questions together results in significantly fewer tokens than answering each question individually.

This observation raises a core research question: can the benefits of resource competition in batch inference be transferred to single-question settings? If so, RLLMs could dynamically adjust their reasoning behaviors, offering concise responses for simple questions while allocating more resources to more complex ones. To this end, we introduce **D**ynamic **R**easoning **Q**uota **A**llocation (**DRQA**), a novel approach that brings the advantages of resource competition into single-question inference, enabling more efficient and adaptive reasoning. Specifically, we first collect diverse reasoning chains under batch inference settings and analyze how the model automatically allocates the length of reasoning chains to problems of varying difficulty in the presence of resource competition. We then construct a preference dataset and introduce a reinforcement learning objective that enables the model to distinguish and learn the advantages of "concise and accurate" reasoning chains over those that are "verbose or incorrect". By indirectly encouraging the model to favor the "concise and accurate" patterns that emerge from batch inference, we enhance its overall reasoning capabilities.

We evaluate the effectiveness of DRQA across a diverse set of reasoning benchmarks, including GSM8K (Cobbe et al., 2021), MATH-500 (Hendrycks et al., 2021), AIME 2024 and 2025 (MAA Committees), AMC (AI-MO, 2024), GPQA-Diamond (Rein et al., 2023) and Live-CodeBench (Jain et al., 2024). Experimental results show that DRQA reduces token usage by over **30%** while consistently maintaining or improving answer accuracy, offering an effective and scalable solution to the overthinking problem. In summary, our main contributions are:

- To the best of our knowledge, we for the first time systematically investigate how *"resource competition pressure"* can enhance the reasoning efficiency of RLLMs during batch inference.
- We propose DRQA, a novel method that transfers this efficiency mechanism to single-question inference by leveraging batch-generated preference data and reinforcement learning. This enables the model to generate *concise answers for simple questions* while maintaining *deep reasoning for complex ones*.
- With extensive experiments, we demonstrate the effectiveness of DRQA compared to existing ones and analyze the results thoroughly.

## 2 RESOURCE COMPETITION DURING BATCH INFERENCE

**Batch Inference Encourages Efficient Reasoning.** As discussed in the introduction, a major challenge for RLLMs is their tendency to overthink, producing unnecessarily long reasoning chains

even for simple questions. To investigate whether batch inference can encourage more efficient reasoning, we conduct a series of controlled experiments. Specifically, we randomly select 500 samples from the DeepScaleR dataset (Luo et al., 2025c) and evaluate several mainstream LLMs under two settings: (i) querying one question at a time (Vanilla), and (ii) querying two questions per prompt (Batch-2). As shown in Table 1, models including DeepSeek-R1 (DeepSeek-AI et al., 2025), Qwen3-32B (think) (Yang et al., 2025a), and Doubao-Seed-1.6 (Seed, 2025) consistently generate shorter outputs in the 'Batch-2' setting, suggesting that batch inference naturally promotes more concise reasoning and that this effect generalizes well across different model architectures.

Table 1: Comparison of average output token lengths across different models under the 'Vanilla' and 'Batch-2' settings.

| Model | Vanilla | Batch-2 |
|---|---|---|
| Deepseek-R1 | 5640.4 | 4035.2 |
| Qwen3-32B (think) | 7761.6 | 5274.7 |
| Doubao-Seed-1.6 | 5288.1 | 3898.2 |

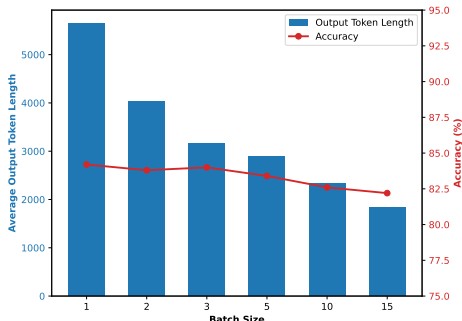

Figure 2: Impact of batch size on output length and accuracy (DeepSeek-R1).

**Scaling Up Batch Size Further Enhances Efficiency.** To further analyze the effect, we vary the batch size using DeepSeek-R1 as a case study, testing batches of **2**, **3**, **5**, **10** and **15** questions. As shown in Figure 2, increasing the batch size leads to a continuous and substantial reduction in the average output length per question. Notably, this compression is achieved with only minimal degradation in answer accuracy, **We hypothesize that this phenomenon stems from an *attention budget* mechanism under context constraints. When processing multiple queries simultaneously, the shared context window acts as a soft bottleneck. To maintain coherence across distinct logical streams, the model is implicitly forced to prioritize high-saliency tokens (core reasoning steps) while suppressing low-information tokens (redundant verbiage). This suggests that 'resource competition' acts as a context-driven information bottleneck, triggering the model's latent capability to compress reasoning without losing semantic integrity.** We refer to this emergent behavior as *resource competition pressure*.

These findings provide compelling empirical evidence that RLLMs are capable of implicit reasoning compression when facing context constraints. The behavior of allocating reasoning resources based on task complexity, without any explicit instruction, points to a promising direction for mitigating the overthinking problem commonly observed in single-question inference. Building on this insight, our work is driven by a central research question: *can we transfer the benefits of resource competition from batch inference to single-question settings?* If so, models could learn to reason adaptively, producing concise answers for simple queries while maintaining sufficient reasoning depth for more complex ones. To this end, we introduce Dynamic Reasoning Quota Allocation (DRQA), detailed in the following section.

## 3 METHODOLOGY

Our goal is to enable RLLMs to assess question complexity and allocate reasoning resources adaptively, even when processing a single query. Ideally, the model should generate short responses for simple problems while preserving sufficient reasoning depth for more challenging ones, thereby improving inference efficiency without compromising answer accuracy. A key challenge in realizing this capability lies in how to effectively transfer "resource competition pressure" from batch inference to single-question settings. We first explore a straightforward solution via supervised fine-tuning (SFT) using batch-generated data. However, this approach revealed inherent limitations in teaching the model to internalize conciseness as a quality criterion. Inspired by recent advancements in Reinforcement Learning with Verifiable Rewards (RLVR) (Lambert et al., 2025; DeepSeek-AI et al., 2025), we introduce Dynamic Reasoning Quota Allocation (DRQA), a reinforcement learning framework that explicitly encourages reasoning that is both accurate and concise. By optimizing an intrinsic reward aligned with these dual objectives, DRQA guides models to dynamically allocate reasoning resources, enabling more efficient and adaptive inference.

Figure 3: The pipeline of Dynamic Reasoning Quota Allocation (DRQA). Batched questions are input to LLM, producing reasoning chains labeled as A/B/C. Reinforcement learning trains the model to prefer concise and accurate reasoning for efficient resource allocation.

Table 2: Single-question evaluation results of Qwen3-8B after SFT with data generated by batch inference. Batch-X denotes fine-tuning with data from batches of X questions, and Vanilla refers to the original model without SFT.

| Method | GSM8K | | Math500 | | AIME2024 | | GPQA-Diamond | | AMC | | AIME2025 | | Overall | |
|---|---|---|---|---|---|---|---|---|---|---|---|---|---|---|
| | Acc | tokens | Acc | tokens | Acc | tokens | Acc | tokens | Acc | tokens | Acc | tokens | Acc | tokens |
| Vanilla | 95.67 | 1878.55 | 96.00 | 5270.58 | 74.67 | 15468.23 | 66.67 | 8685.21 | 97.50 | 8608.85 | 63.33 | 18058.65 | 82.31 | 9661.68 |
| Batch-2 | 96.67 | 575.64 | 95.00 | 2359.21 | 57.33 | 11100.55 | 53.54 | 6874.42 | 90.00 | 4136.58 | 45.33 | 13130.95 | 72.98 | 6362.89 |
| Batch-3 | 93.33 | 437.23 | 82.67 | 1593.53 | 26.00 | 5685.36 | 55.56 | 3555.65 | 77.50 | 4098.10 | 28.00 | 7400.53 | 60.51 | 3795.07 |
| Batch-5 | 93.33 | 336.81 | 69.67 | 434.50 | 9.33 | 2486.77 | 46.46 | 1190.23 | 42.50 | 922.25 | 7.33 | 2365.41 | 44.77 | 1289.33 |

## 3.1 SUPERVISED FINE-TUNING WITH BATCH DATA

Our initial approach to transferring the benefits of resource competition into single-question inference is based on imitation learning, where we apply supervised fine-tuning (SFT) to mimic the efficient reasoning patterns exhibited by models during batch inference.

**Method**  We use DeepSeek-R1 (DeepSeek-AI et al., 2025) to perform batch inference over multiple questions sampled from DeepScaleR (Luo et al., 2025c) and collect the generated responses, which are consistently more concise than those from single-question inference. Based on these results, we construct a dataset of "question–concise answer" pairs and apply full-parameter SFT on a Qwen3-8B (Yang et al., 2025a), with the goal of teaching it to generate similarly concise responses in single-question scenarios.

**Experimental Results and Analysis**  We evaluate the fine-tuned models on a comprehensive set of reasoning benchmarks, including GSM8K (Cobbe et al., 2021), MATH-500 (Hendrycks et al., 2021), AIME 2024 (MAA Committees), GPQA-Diamond (Rein et al., 2023), AMC (AI-MO, 2024), and AIME 2025 (MAA Committees). The results shown in Table 2 indicate that SFT does lead to substantial reductions in output length. For example, on GSM8K, the average response length drops from 1878.55 to 575.64 tokens, a 69.36% reduction, demonstrating that overthinking is mitigated to some extent.

However, the efficiency gains come at a considerable cost to accuracy, particularly on more challenging tasks. As shown in Table 2, Models fine-tuned with two-question batch data show a slight accuracy increase from 95.67% to 96.67% on GSM8K, while on MATH-500 accuracy drops from 96.00% to 95.00%, a decrease of 1.00% compared to vanilla prompting. More notably, the performance degradation becomes increasingly severe with higher batch sizes and task complexity. On AIME 2024, accuracy falls from 74.67% (Vanilla) to 57.33% (Batch-2), 26.00% (Batch-3), and just 9.33% (Batch-5). These results suggest the emergence of catastrophic forgetting (Luo et al., 2025d): in attempting to mimic the surface-level conciseness of batch responses, the model compromises its ability to perform the deeper, more nuanced reasoning necessary for solving complex problems.

In summary, while supervised fine-tuning with batch data effectively mitigates overthinking and improves inference efficiency, it comes at the cost of reasoning accuracy, especially on complex tasks,

highlighting its limitations for real-world deployment. These shortcomings underscore the need for a more principled solution that can balance conciseness with reasoning depth, which motivates our proposed method: Dynamic Reasoning Quota Allocation (DRQA).

## 3.2 DYNAMIC REASONING QUOTA ALLOCATION

Rather than imitating outputs from batch inference, we aim to endow the model with an intrinsic ability to evaluate and generate reasoning chains that are both accurate and concise. To this end, we propose Dynamic Reasoning Quota Allocation (DRQA), a reinforcement learning framework that enables RLLMs to dynamically allocate reasoning resources in single-question inference.

**Core Idea**   The core idea of DRQA is to enhance the model's intrinsic reasoning capabilities by equipping it with the ability to evaluate the quality of its own reasoning chains. Specifically, the model is trained to make two key judgments: (i) whether a given reasoning chain is logically correct, and (ii) if correct, whether it is unnecessarily verbose. *By developing this self-evaluation ability, the model learns to strike a balance between accuracy and conciseness during generation, effectively realizing adaptive resource allocation.*

**Preference Data Construction**   To train this evaluation ability, we construct a preference dataset consisting of multiple-choice question-answering samples. Each sample contains a question, a model-generated chain of thought (CoT), and three evaluation options that reflect different levels of reasoning quality:

- **A**: The reasoning process is correct, but I think there is a simpler and quicker way to approach it.
- **B**: The reasoning process is correct, and I believe the thinking is thorough and concise.
- **C**: The reasoning process is wrong.

The dataset construction process involves three key steps. First, for ease of evaluation, we select all questions in the DeepScaleR (Luo et al., 2025c) dataset whose answers are numbers of various types, resulting in approximately 30,000 samples. Second, for each question, we generate two types of reasoning chains using DeepSeek-R1 (DeepSeek-AI et al., 2025): (1) *vanilla CoTs* obtained by prompting the model with individual questions, and (2) *batch CoTs* generated by prompting the model with batched questions, followed by extracting the corresponding reasoning chain for each question. Finally, we assign labels based on reasoning correctness and conciseness: for vanilla CoTs, we label **A** if the reasoning is correct, and **C** if incorrect; for batch CoTs, we label **B** if the reasoning is correct, and **C** if incorrect. This labeling scheme enables the model to learn nuanced distinctions between correct-but-verbose reasoning (option A), correct-and-concise reasoning (option B), and incorrect reasoning (option C), thereby developing a clearer understanding of what constitutes a high-quality reasoning chain.

**Reinforcement Learning Framework**   We use Group Relative Policy Optimization (GRPO) (Shao et al., 2024) to train the model to accurately classify each reasoning chain as A, B, or C, thus encouraging concise and accurate reasoning. Formally, the GRPO objective is defined as maximizing the likelihood of selecting the correct evaluation label:

$$\mathcal{L}_{\text{GRPO}}(\theta) = \mathbb{E}_{\tau \sim \mathcal{D}} \left[ \sum_{a \in \mathcal{G}} \log \pi_\theta(a \mid s) \ \hat{A}(a, s, a^*) - \beta \ \text{KL}(\pi_\theta \,\|\, \pi_{\text{old}}) \right] \tag{1}$$

where $\tau \sim \mathcal{D}$ denotes a sample from the dataset, with state $s$ representing the question, reasoning chain, and multiple-choice options (A, B, C); $a^*$ is the ground-truth label; $\mathcal{G} = \{A, B, C\}$ is the set of actions; $\hat{A}(a, s, a^*)$ is the relative advantage estimate, positive if $a = a^*$ and negative otherwise; $\text{KL}(\pi_\theta \| \pi_{\text{old}})$ is the KL divergence between the current and old policies, constrains the policy update; and $\beta$ is a regularization coefficient balancing learning efficiency and policy stability. This training objective encourages the model to assign higher probabilities to correct judgments while mitigating the risk of catastrophic forgetting caused by over-updating, a common issue encountered in SFT. As a result, the model gradually internalizes a preference for reasoning chains that are both correct and concise.

**Summary**   DRQA enables the model to move beyond surface-level imitation and develop an intrinsic, reward-driven preference for high-quality reasoning. By balancing accuracy and conciseness, the model learns to allocate reasoning resources more effectively, addressing the limitations of SFT and supporting more efficient and adaptive inference in single-question settings.

# 4 EXPERIMENTS

In this section, we systematically evaluate the performance of the proposed DRQA algorithm, focusing on its ability to balance reasoning accuracy and efficiency. We compare DRQA against a range of strong baselines and provide an in-depth analysis of the results.

## 4.1 EXPERIMENTAL SETUP

**Models**   We evaluate all methods using three widely adopted distilled models: DeepSeek-R1-Distill-Qwen-1.5B, DeepSeek-R1-Distill-Qwen-7B, and DeepSeek-R1-Distill-Llama-8B. All models are derived from the more powerful DeepSeek-R1 (DeepSeek-AI et al., 2025) through large-scale distillation, offering a favorable trade-off between computational efficiency and reasoning capability.

**Datasets**   For training, we use the dataset described in Section 3.2, constructed by performing batch inference with DeepSeek-R1 on the DeepScaleR (Luo et al., 2025c) training set. This process yields over 50,000 multiple-choice examples annotated with reasoning quality labels.

**Baselines**   To assess the effectiveness of DRQA, we compare it against a comprehensive set of strong baselines approaches (refer to Appendix A for detailed descriptions of the baselines). All baselines are either publicly released or carefully reproduced according to their original protocols. **We note that these baselines span different paradigms, methods like AutoL2S train base models for efficient reasoning, while others and our DRQA focus on compressing or adapting powerful existing RLLMs. We include this diverse set to map the full landscape of efficient reasoning techniques.**

**Evaluation**   We evaluate the performance of different methods across a diverse set of benchmarks. For mathematical reasoning, we include GSM8K (Cobbe et al., 2021), MATH-500 (Hendrycks et al., 2021), AIME 2024 and 2025 (MAA Committees), and AMC 2023 (AI-MO, 2024). For domain-specific scientific reasoning, we use the high-quality GPQA-diamond subset (Rein et al., 2023). Detailed descriptions of these datasets are provided in Appendix B. We use both accuracy and response length as evaluation metrics and report the average performance across all test sets. For the AIME datasets, which contain only 30 questions each, we repeatedly sample 5 responses for each case and report the average results to ensure more stable and reliable evaluation.

All models are evaluated using a unified inference configuration to ensure fair comparison. Experiments are conducted with the vLLM framework on a computing cluster equipped with eight A800 (40GB) GPUs. The inference parameters are set to a temperature of $0.6$ and a maximum generation length of 32K tokens.

**Training Details**   We use verl (Sheng et al., 2024) as the training framework. We set the batch size to 256, the number of rollouts to 16, the learning rate to $1 \times 10^{-6}$, and the maximum response length to 16K tokens. The model is trained for one epoch, consisting of 204 steps in total.

## 4.2 MAIN RESULTS

As shown in Table 3, DRQA demonstrates clear superiority in both answer accuracy and response efficiency across all mathematical benchmarks. For example, on GSM8K with the 1.5B model, DRQA achieves an accuracy of 86.67%, outperforming the vanilla baseline by 2 percentage points, while reducing average token usage from 1928.96 to 1427.63, a 25.9% reduction. Similar patterns are observed on more challenging datasets such as AIME 2024 and MATH-500, where DRQA maintains high accuracy while significantly reducing output length. These results highlight DRQA's effectiveness in dynamically allocating reasoning resources, enabling it to strike a favorable balance between accuracy and efficiency across tasks of varying difficulties. Moreover, DRQA demonstrates

Table 3: Performance of different methods using three RLLMs: DeepSeek-R1-Distill-Qwen-1.5B, DeepSeek-R1-Distill-Qwen-7B, and DeepSeek-R1-Distill-Llama-8B. DRQA achieves competitive or superior accuracy while greatly reducing token usage across all datasets and model variants, striking an excellent balance between performance and efficiency.

| Method | GSM8K | | MATH-500 | | AIME 2024 | | GPQA-Diamond | | AMC 2023 | | AIME 2025 | | Overall | |
|---|---|---|---|---|---|---|---|---|---|---|---|---|---|---|
| | Acc | Tokens | Acc | Tokens | Acc | Tokens | Acc | Tokens | Acc | Tokens | Acc | Tokens | $Acc_{All}$ | $Tokens_{All}$ |
| **DeepSeek-R1-Distill-Qwen-1.5B** | | | | | | | | | | | | | | |
| Vanilla | 84.67% | 1928.96 | 83.33% | 5536.14 | 28.67% | 14394.61 | 30.84% | 14731.59 | 72.50% | 8830.10 | 23.67% | 15323.3 | 53.95% | 10124.12 |
| O1-Pruner | 74.80% | 458 | 82.20% | 3212 | 28.90% | 10361 | - | - | - | - | - | - | – | – |
| DAST | 77.20% | 586 | 83.00% | 2428 | 26.90% | 7745 | - | - | - | - | - | - | – | – |
| ShortBetter | 63.67% | **107.86** | 60.33% | **1186.27** | 11.33% | **2935.68** | 21.72% | **1433.95** | 57.50% | **1260.43** | 12.67% | **3326.22** | 37.87%$_{-16.08}$ | **1708.40**$_{-83.13\%}$ |
| AdaptThink | 86.00% | 324.26 | 83.67% | 1244.98 | 29.33% | 7044.06 | 29.80% | 4744.23 | 72.50% | 2441.45 | 24.67% | 7490.79 | 54.33%$_{+0.38}$ | 3881.63$_{-61.66\%}$ |
| GRPO | **87.33%** | 1691.19 | 84.67% | 5743.01 | **32.67%** | 15017.54 | 27.78% | 13809.53 | **77.50%** | 9378.21 | 24.00% | 13082.98 | 55.66%$_{+1.71}$ | 9787.08$_{-3.33\%}$ |
| GRPO+Length Penalty | 86.00% | 722.34 | 84.67% | 2479.14 | 24.67% | 9011.46 | 26.76% | 6148.50 | 67.50% | 3130.51 | 22.00% | 9782.34 | 51.93%$_{-2.01}$ | 5212.38$_{-48.52\%}$ |
| SFT | 81.67% | 2296.54 | 80.33% | 5465.95 | 25.33% | 21337.44 | 27.27% | 18540.94 | 65.00% | 8806.48 | 19.33% | 20258.62 | 49.82%$_{-4.13}$ | 12784.36$_{-26.28\%}$ |
| DRQA(our) | 86.67% | 1427.63 | **84.67%** | 3488.08 | 32.00% | 11008.31 | **31.81%** | 9148.83 | 75.00% | 5355.03 | 24.00% | 10382.12 | **55.69%**$_{+1.74}$ | 6801.67$_{-32.82\%}$ |
| **DeepSeek-R1-Distill-Qwen-7B** | | | | | | | | | | | | | | |
| Vanilla | 91.33% | 1735.5 | 90.40% | 5099.95 | 53.33% | 13712.6 | 48.98% | 13313.92 | 90.00% | 6349.53 | 40.00% | 14248.11 | 69.01% | 9076.60 |
| DAST | 86.70% | 459 | 89.60% | 2162 | 45.60% | 7578 | - | - | - | - | - | - | – | – |
| O1-Pruner | 87.60% | 428 | 86.60% | 2534 | 49.20% | 9719 | - | - | - | - | - | - | – | – |
| Dynasor-CoT | 89.60% | 1285 | 89.00% | 2971 | 46.70% | 12695 | 30.50% | 7639 | 85.00% | 5980 | - | - | – | – |
| DEER | 90.60% | 6917 | 89.80% | 2143 | 49.20% | 9839 | 31.30% | 5469 | 85.00% | 4451 | - | - | – | – |
| ShortBetter | 70.00% | **112.86** | 68.00% | **623.44** | 41.33% | **5005.96** | 43.43% | **1811.43** | 57.50% | **1567.50** | 30.67% | 5393.96 | 51.82%$_{-17.19}$ | **2419.19**$_{-73.35\%}$ |
| AdaptThink | 89.67% | 296.94 | 91.67% | 1839.59 | 54.00% | 9894.05 | 51.52% | 7128.95 | 87.50% | 3287.95 | 39.33% | 12454.59 | 68.95%$_{-0.06}$ | 5817.01$_{-35.91\%}$ |
| AutoL2S | 93.33% | 444.8 | 83.33% | 3113.93 | 40.67% | 6499.32 | 45.39% | 2553.01 | 85.00% | 2613.05 | 31.33% | **3669.53** | 63.18%$_{-5.84}$ | 3148.94$_{-65.31\%}$ |
| GRPO | **93.67%** | 1524.24 | **92.00%** | 4532.21 | **54.67%** | 12013.92 | 47.47% | 12124.10 | 87.50% | 5130.13 | **41.33%** | 12192.12 | 69.44%$_{+0.43}$ | 7919.45$_{-12.75\%}$ |
| GRPO+Length Penalty | 91.33% | 876.25 | 91.33% | 2751.13 | 52.00% | 7213.11 | 45.96% | 7124 | **92.50%** | 3256.02 | 39.67% | 6058.40 | 68.80%$_{-0.21}$ | 4546.49$_{-49.91\%}$ |
| SFT | 92.33% | 1317.85 | 92.00% | 3824.43 | 44.67% | 14903.82 | 46.97% | 12385.43 | 77.50% | 5519.55 | 32.00% | 13931.80 | 64.25%$_{-4.76}$ | 8647.15$_{-4.73\%}$ |
| DRQA(our) | 92.67% | 1324.24 | 91.40% | 3902.74 | 54.67% | 10007.18 | **49.50%** | 8988.50 | 92.50% | 4463.03 | 40.67% | 9545.44 | **70.24%**$_{+1.23}$ | 6371.85$_{-29.80\%}$ |
| **DeepSeek-R1-Distill-Llama-8B** | | | | | | | | | | | | | | |
| Vanilla | 91.67% | 1829.12 | 90.00% | 5417.41 | 49.33% | 13585.12 | 48.98% | 11845.27 | 87.50% | 7177.73 | 38.67% | 14260.26 | 67.69% | 9019.15 |
| GRPO | 92.33% | 1605.94 | **91.67%** | 4812.02 | 50.67% | 12897.09 | 46.46% | 9869.20 | 90.00% | 7600.58 | 39.33% | 12204.58 | 68.41%$_{+0.72}$ | 8164.90$_{-9.47\%}$ |
| GRPO+Length Penalty | 91.67% | **875.66** | 91.33% | 2753.43 | 48.00% | **7192.28** | 45.96% | 7055.54 | 90.00% | **3236.22** | 38.00% | **8040.74** | 67.49%$_{-0.20}$ | **4858.98**$_{-46.13\%}$ |
| SFT | 90.67% | 1315.83 | 90.00% | 3825.52 | 44.67% | 14881.25 | 44.95% | 10897.06 | 75.00% | 5509.82 | 32.67% | 13915.29 | 62.99%$_{-4.70}$ | 8390.80$_{-6.97\%}$ |
| DRQA(our) | **93.00%** | 1594.70 | 91.33% | 4180.83 | **50.67%** | 9940.46 | **50.00%** | 8986.63 | **92.50%** | 4463.43 | **39.33%** | 9542.11 | **69.47%**$_{+1.78}$ | 6451.36$_{-28.47\%}$ |

strong generalization on out-of-distribution (OOD) benchmarks, as evidenced by its performance on GPQA-Diamond.

We also compare DRQA with aggressive compression methods such as ShorterBetter (Yi et al., 2025) and DAST (Shen et al., 2025), which can reduce output length even further, for example, generating outputs as short as 107.86 tokens on GSM8K. However, these methods often suffer from severe accuracy degradation, with performance drops exceeding 20 percentage points in some cases. This highlights a key limitation of methods that rely solely on length-based reward signals: they tend to compromise the logical integrity of reasoning chains, limiting their practical applicability.

Notably, DRQA remains highly effective on larger models. On GSM8K with the 7B model, DRQA improves accuracy by 1.34% over the baseline while reducing token usage by 23.6%. On Llama-8B, DRQA achieves a 1.78% accuracy gain while cutting token usage by 28.47%, highlighting its ability to enhance performance and efficiency at larger model scales. Across all benchmarks, it consistently achieves the most favorable trade-off between accuracy and output efficiency. Compared to strong baselines such as DAST (Shen et al., 2025), O1-Pruner (Luo et al., 2025b), Dynasor-CoT (Fu et al., 2025), and DEER (Xia et al., 2024), DRQA not only matches or surpasses them in length reduction but, more importantly, maintains state-of-the-art reasoning accuracy.

Overall, DRQA achieves an average accuracy improvement of 1.58 percentage points and an average token usage reduction of 30.4% across all evaluated benchmarks and all three model variants. These results provide compelling evidence that DRQA effectively transfers the benefits of "resource competition pressure" from batch inference to single-question settings, establishing a strong foundation for the efficient and scalable deployment of RLLMs.

## 4.3 GENERALIZATION TO CODE GENERATION

We further assess DRQA on the LiveCodeBench benchmark (Jain et al., 2024), a contamination-free suite of code-related tasks collected from competitive programming platforms. Our evaluation uses 342 newly released Python problems spanning September 2024 and April 2025. As shown in Table 4, DRQA consistently reduces token usage by about 23%–29% across all three model sizes, while also improving accuracy. For example, on *DeepSeek-R1-Distill-Qwen-7B*, DRQA shortens outputs from 8724.27 to 6648.77 tokens (-23.79%) and improves accuracy by 1.75%, demonstrating its strong generalizability to the code generation domain.

Table 4: Performance on LiveCodeBench. **Include GRPO and GRPO+Length Penalty as additional baselines. DRQA achieves the best accuracy across all model sizes while significantly reducing token usage. While GRPO+Length Penalty achieves extreme shortness, it suffers from accuracy degradation, whereas DRQA maintains a superior balance.**

| Method | DeepSeek-R1-Distill-Qwen-1.5B | | DeepSeek-R1-Distill-Qwen-7B | | DeepSeek-R1-Distill-Llama-8B | |
|---|---|---|---|---|---|---|
| | Acc | Tokens | Acc | Tokens | Acc | Tokens |
| *Vanilla* | 13.16% | 11261.72 | 30.70% | 8724.27 | 31.87% | 9012.31 |
| ***GRPO*** | **13.45%** | **10845.20** | **31.29%** | **8412.33** | **32.16%** | **8640.12** |
| ***GRPO+Length Penalty*** | **11.99%** | **6514.50** | **29.24%** | **5920.45** | **30.12%** | **5890.76** |
| *DRQA(our)* | **13.74%** | 8124.20 | **32.45%** | 6648.77 | **32.75%** | 6426.68 |

Table 5: Ablation experiments across different training paradigms.

| Method | GSM8K | | MATH-500 | | AIME 2024 | | Overall | |
|---|---|---|---|---|---|---|---|---|
| | Acc | tokens | Acc | tokens | Acc | tokens | Acc | tokens |
| *Vanilla* | 91.33% | 1735.5 | 90.40% | 5099.95 | 53.33% | 13712.6 | 78.35% | 6849.35 |
| *DRQA (Batch-2)* | 92.67% | 1324.24 | 91.33% | 3902.74 | 54.67% | 10007.18 | **79.58%**$_{+1.23}$ | 5078.05$_{-25.86\%}$ |
| *DRQA (Batch-3)* | 91.67% | 1212.59 | 90.20% | 3311.20 | 53.33% | 8805.24 | 78.40%$_{+0.05}$ | 4443.01$_{-35.13\%}$ |
| *DRQA (Batch-5)* | 90.67% | 1158.88 | 89.80% | 2675.81 | 49.33% | 7366.80 | 76.60%$_{-1.75}$ | **3733.83**$_{-45.49\%}$ |
| *Qwen2.5-7B Data + RL* | 90.00% | 1434.65 | 89.60% | 3313.12 | 50.67% | 12190.59 | 76.76%$_{-1.60}$ | 5646.12$_{-17.57\%}$ |
| *Batch-2 Data + CFT* | 89.67% | 1361.00 | 88.20% | 3973.54 | 49.66% | 10012.55 | 75.84%$_{-2.51}$ | 5115.70$_{-25.31\%}$ |

## 4.4 ABLATION STUDY

To thoroughly assess the contribution of each core component in DRQA, we conduct a series of ablation studies that isolate the effects of different training paradigms and input conciseness on reasoning performance and efficiency. All experiments are performed using the same benchmark datasets, evaluation metrics, and base model (DeepSeek-R1-Distill-Qwen-7B) as in the main study, with consistent inference configurations to ensure fair comparison.

**Effect of Batch Size in DRQA Data Construction.** We investigate the impact of different batch sizes on model performance. Specifically, we construct preference datasets by prompting DeepSeek-R1 with batches of 2, 3, or 5 questions, then splitting the outputs into individual reasoning chains for downstream RL training. This design allows us to analyze how increasing levels of resource competition influence both answer accuracy and response efficiency within the DRQA framework.

**Replacing Batch Reasoning Data with Qwen2.5-7B Concise Chains** To evaluate the importance of batch-induced resource competition, we consider an alternative setting where the preference dataset is constructed using concise reasoning chains generated directly by Qwen2.5-7B (Qwen et al., 2025), without leveraging batch inference. This comparison allows us to disentangle the effects of resource-driven compression from those achieved solely through the model's inherent ability to generate concise outputs.

**Critique Fine-Tuning with Preference data** Beyond reinforcement learning, we also evaluate the Critique Fine-Tuning (CFT) paradigm (Wang et al., 2025) as an alternative training strategy, applying it to the preference data we constructed.

## 4.4.1 RESULTS AND ANALYSIS

Table 5 presents the results of our ablation study. As batch size increases, the model produces increasingly concise outputs, with token usage reduced by up to 45% for larger batches. However, this efficiency gain comes at the cost of declining accuracy, highlighting a trade-off between efficiency and correctness. Notably, a batch size of 2 achieves the best balance, improving accuracy while significantly reduc-

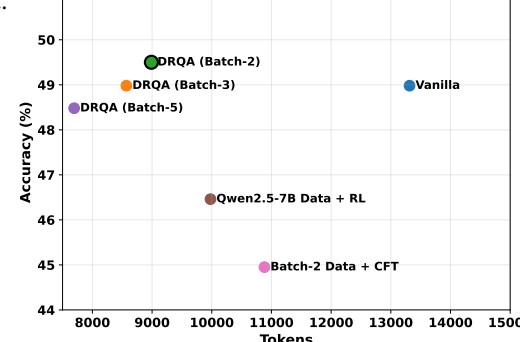

Figure 4: The efficiency-accuracy trade-off on GPQA-diamond for DRQA and ablation variants.

ing token consumption compared to the vanilla
baseline.

When compared to concise reasoning chains generated directly by Qwen2.5-7B Qwen et al. (2025) without batch inference, we observe that only batch-induced compression achieves both high efficiency and strong accuracy. Similarly, while Critique Fine-Tuning helps reduce output length, it leads to a notable accuracy drop, underscoring the importance of reinforcement learning for preserving reasoning quality. Figure 4 further supports these insights, showing that DRQA achieves the best overall trade-off on the OOD dataset GPQA-Diamond, highlighting its robustness across both in-distribution and out-of-distribution scenarios.

## 5 RELATED WORK

### 5.1 REASONING LARGE LANGUAGE MODELS

Recent advances in reasoning large language models (RLLMs), such as OpenAI-O3 (OpenAI, 2025), Deepseek-R1 (DeepSeek-AI et al., 2025), and QwQ (Team, 2025) leverage chain-of-thought (Wei et al., 2023) for step-by-step reasoning, achieving state-of-the-art performance across tasks including mathematical reasoning, coding, and complex question answering. CoT allows these models to leverage inference-time scaling by generating multiple reasoning steps that explore alternative solution paths, thereby significantly enhancing accuracy over single-pass generation. To further improve correctness, a variety of methods have been proposed, including self-consistency (Wang et al., 2023), beam search (Yao et al., 2023), and reinforcement learning-based post-training (DeepSeek-AI et al., 2025), which encourage iterative self-reflection and help reduce logical errors. Additional search-based approaches, such as Monte Carlo Tree Search (MCTS) (Gao et al., 2024), have been employed to expand the scope of exploration in complex problem-solving scenarios. Our work focuses on further improving the efficiency of such reasoning models.

### 5.2 EFFICIENT REASONING

Reasoning efficiency in RLLMs (Qu et al., 2025; Sui et al., 2025) refers to balancing task quality and computational cost. Models like OpenAI-O3 (OpenAI, 2025) and DeepSeek-R1 (DeepSeek-AI et al., 2025) often generate too long and redundant reasoning chains, over explaining simple problems while sometimes offering shallow reasoning for complex ones. Main approaches for improving efficiency include:

- **Inference time control**: Methods such as TALE (Han et al., 2025), DEER (Yang et al., 2025b) apply token budgets or early exit strategies inspired by dual-system theory.
- **Chain compression and supervised tuning**: TokenSkip (Xia et al., 2025), CoT-Valve (Ma et al., 2025), and AutoL2S (Luo et al., 2025a) use supervised fine-tuning or distillation to shorten reasoning chains, often improving conciseness but sometimes at the expense of complex reasoning.
- **Reinforcement learning approaches**: DAST (Shen et al., 2025), O1-Pruner (Luo et al., 2025b), and S-GRPO (Dai et al., 2025) introduce reward functions to penalize lengthy outputs and promote token efficiency, supporting adaptive reasoning with little loss of accuracy.

These methods largely depend on fixed budgets or hand crafted rewards. Our DRQA instead transfers the "resource competition pressure' observed in batch inference to single-question settings, enabling models to automatically adjust reasoning length according to problem complexity, providing brief responses for simple questions and detailed explanations for challenging ones without manual constraints.

### 5.3 BATCH PROMPTING AND RESOURCE COMPETITION

Batch prompting (Cheng et al., 2023) was originally proposed to improve inference throughput and reduce API costs by grouping multiple samples into a single prompt, allowing shared instructions and few-shot exemplars. Subsequent works have optimized this paradigm to address stability and performance issues:

- **Robustness and ordering**: Lin et al. (2024) introduced permutation and self-consistency mechanisms to mitigate position bias and performance degradation often observed in batched inputs.
- **Demonstration utilization**: Feng et al. (2024) proposed "Auto-Demo Prompting," which leverages generated outputs from earlier queries within a batch as demonstrations for subsequent ones to enhance performance.

These methods primarily view batching as an **inference-time optimization** technique to amortize computational overhead across multiple queries. Our work draws inspiration from the context constraints observed in these studies but fundamentally differs in objective and mechanism. While prior works aim to maintain performance while maximizing batch size for throughput, DRQA identifies that the "resource competition" inherent in batching naturally suppresses redundancy in reasoning chains. Instead of using batching solely for inference speedup, we leverage it as a **data generation mechanism** for training. By teaching the model to internalize this concise reasoning pattern via reinforcement learning, DRQA transfers the efficiency benefits of batching to **single-question inference**, enabling dynamic reasoning quota allocation without requiring batched inputs at test time.

## 6 Conclusion

This paper introduces Dynamic Reasoning Quota Allocation (DRQA), a novel approach aimed at addressing the overthinking problem in reasoning large language models (RLLMs). Motivated by the observation that resource competition pressure in batch inference naturally encourages efficient reasoning, DRQA leverages batch-generated data and reinforcement learning to transfer the benefits of resource competition from batch inference to single-question scenarios. Specifically, the model is trained to develop an internal preference for reasoning processes that balance conciseness with accuracy, allowing it to produce short answers for straightforward questions while preserving adequate reasoning depth when tackling more complex ones. Extensive experimental results and analysis show that DRQA significantly reduces token consumption while maintaining, or even improving, accuracy. By effectively alleviating overthinking, DRQA offers a new direction for more efficient and scalable deployment of RLLMs.

## 7 Ethics Statement

This work adheres to the ICLR Code of Ethics and follows responsible research practices. Our study focuses on improving the efficiency of reasoning in large language models (LLMs) by mitigating overthinking through the proposed Dynamic Reasoning Quota Allocation (DRQA) framework.

The research does **not** involve human subjects, personal data, or sensitive information. All datasets used (e.g., GSM8K, MATH-500, AIME, AMC, GPQA-Diamond, LiveCodeBench, DeepScaleR) are publicly available, widely adopted in the AI community, and contain no personally identifiable information; data usage strictly complies with their respective licenses.

Our experiments are performed entirely in silico, and the outputs are automatically generated by models without human intervention. The proposed method is designed to **reduce computational cost and energy usage** by shortening reasoning chains for simple queries while retaining depth for complex ones, thereby contributing positively to environmental sustainability.

We have considered possible risks, including unintended accuracy degradation in complex tasks or misuse of the system for harmful automated decision making. Mitigation strategies include thorough benchmark evaluation across diverse domains, public release of methodology for reproducibility, and clear documentation of model limitations.

The study promotes fairness and avoids discrimination by focusing on general-purpose mathematical and scientific datasets with balanced coverage; no content in our dataset or method is intended to target or disadvantage any demographic group.

All code, data handling, and result reporting are conducted with scientific integrity, transparency, and reproducibility in mind. We believe this work supports the responsible advancement of AI that serves the public good, efficiency, and well-being.

## 8 REPRODUCIBILITY STATEMENT

We have implemented several measures to ensure that all reported results are fully reproducible. A comprehensive description of the proposed DRQA algorithm, including the procedures for batch data collection, preference dataset construction, and the reinforcement learning framework, is provided in 3.2. The precise model variants, datasets, and evaluation metrics are specified in section 4 and in Appendix B. Inference hyperparameters and training configurations, such as the optimizer type, batch size, learning rate, number of rollouts, and maximum generation length, are detailed in section 4.1.

Two code modules are provided as supplementary material. The first module, `verl/`, contains the implementation of DRQA training using Group Relative Policy Optimization (GRPO). The second module, `eval/`, contains scripts for generating training data via batch inference, conducting evaluations on all benchmarks, and reproducing the tables and figures reported in the paper.

The supplementary code package includes complete, end-to-end instructions for dataset preprocessing, model training, and evaluation. All datasets used in training and evaluation, including GSM8K, MATH-500, AIME 2024, AIME 2025, AMC 2023, GPQA-Diamond, DeepScaleR, and LiveCodeBench, are publicly available. Our preprocessing steps are documented in the supplementary material.

With the provided source code, configuration files, and dataset references, independent researchers can exactly reproduce our experiments and validate all results under identical conditions.

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

# A  BASELINE METHODS

We consider the following baseline methods in our experiments:

- **GRPO:** We train a model on the DeepScaleR (Luo et al., 2025c) dataset using the Group Relative Policy Optimization algorithm, where only answer correctness is used as the reward signal.
- **GRPO+Length Penalty:** This variant further introduces a length penalty to the reward design: for correct answers, shorter responses yield higher rewards, while for incorrect answers, longer responses incur greater penalties. This encourages the model to produce concise and accurate reasoning.
- **SFT (Supervised Fine-Tuning):** We perform full-parameter supervised fine-tuning on the model using question-answer pairs generated via batch inference of Deepseek-R1 on the Deep-ScaleR (Luo et al., 2025c) dataset.
- **AdaptThink** (Zhang et al., 2025): This approach encourages adaptive selection between direct answer and step-by-step reasoning (Chain-of-Thought) based on question difficulty. Training objectives and sample balancing enable the model to flexibly explore both thinking modes, improving reasoning efficiency and performance.
- **AutoL2S** (Luo et al., 2025a): A dynamic, model-agnostic framework that annotates each question with both long and short Chain-of-Thought (CoT) solutions. By marking simple questions with `<EASY>`, the model is trained to automatically select concise CoT for simple problems and detailed reasoning for complex ones. **Distinct from methods that distill reasoning models, AutoL2S is typically trained starting from a base model (e.g., Qwen2.5-7B) to achieve efficient reasoning.**
- **DAST** (Shen et al., 2025): DAST explicitly quantifies problem difficulty via a token length budget and employs a reward that penalizes redundant reasoning on simple problems while encouraging extensive CoT for difficult ones. This preference data is optimized via SimPO, enabling efficient dynamic control over reasoning path length.
- **O1-Pruner** Luo et al. (2025b): Based on reinforcement learning, this method rewards shorter CoT traces without compromising accuracy. It employs an offline PPO-like procedure to prune redundant reasoning while preserving or even improving correctness.
- **ShorterBetter** (Yi et al., 2025): This RL-based approach defines the optimal length for each question as the shortest possible correct response and leverages this dynamic signal as a reward for GRPO-based training, guiding the model toward concise yet accurate answers.
- **Dynasor-CoT** (Fu et al., 2025): Without extra training, this method dynamically truncates reasoning by probing intermediate answers, monitoring consistency, and detecting hesitancy tokens. This yields substantial token savings while preserving accuracy.
- **DEER** (Xia et al., 2024): DEER employs a dynamic early-exit mechanism by monitoring reasoning transitions (such as "Wait") to induce trial answers. Decisions to terminate CoT generation are based on confidence estimation, reducing reasoning length without additional training.

All baseline models are tested under identical inference configurations and on the same benchmark datasets to guarantee fair and reliable comparison. For each baseline, we use either the officially released model or reproduce the method using released data and code.

# B  DATASET DETAILS

**Mathematical Reasoning Datasets**

- **GSM8K**: This dataset contains 8,500 English elementary school single-step math reasoning questions. It serves as one of the mainstream benchmarks for evaluating the math reasoning abilities of large language models, focusing on basic arithmetic reasoning skills.
- **MATH-500**: Includes 500 medium-difficulty mathematical problems covering algebra, geometry, number theory, and other areas, designed to test the model's comprehensive mathematical reasoning ability.
- **AIME 2024/2025**: Originating from the American Invitational Mathematics Examination 2024 and 2025, each set contains 30 high-difficulty math questions, mainly assessing complex mathematical reasoning and problem-solving skills.

- **AMC 2023**: 40 questions from the American Mathematics Competitions (AMC), covering middle to high school levels, examining fundamental and advanced mathematics knowledge and problem solving abilities.

**Scientific Reasoning Dataset** To evaluate model reasoning performance in other domains, we use the high-quality GPQA-diamond subset from the GPQA dataset. GPQA-diamond is a refined version of GPQA, focusing on challenging, high-quality scientific domain questions and designed to provide a comprehensive assessment of scientific understanding and reasoning ability.

## C  PROMPT TEMPLATE

---
**Prompt for Batch Inference**

Please answer the following math problems in order and summarize all answers at the end: Your response should be in the following format:

```
[Solution Process]
Provide a detailed solution for each problem...

[Final Answer]
1. \\boxed{{Answer1}}
2. \\boxed{{Answer2}}
...
n. \\boxed{{Answern}}
```

Below is the list of questions:

```
{numbered_questions}
```
---

---
**Prompt for Evaluation**

```
{origin_question}\n\n
Please reason step by step, and put your final answer
within \\boxed{}.
```
---

## D  THE USE OF LARGE LANGUAGE MODELS (LLMS)

In the preparation of this manuscript, we made limited use of publicly available large language models (specifically GPT-5) solely to improve the clarity, grammar, and readability of the English text. The LLM was **not** used for research ideation, hypothesis generation, experiment design, data analysis, or the production of any scientific content. All technical ideas, experiments, analyzes, and conclusions presented in this paper are completely authored by the listed human authors. The authors have carefully reviewed and verified the accuracy of all LLM-assisted edits and take full responsibility for the final content.

## E  QUALITATIVE EXAMPLES

**In this section, we provide qualitative comparisons between the baseline *DeepSeek-R1-Distill-Qwen-7B* (Vanilla) and our *DRQA-7B* model. We explicitly denote the total token consumption for each response.**

## E.1  CASE STUDY 1: GSM8K

---

**Question**

**A church has 120 members. 40% are adults. The rest are children. How many more children are there than adults?**

---

**DeepSeek-R1-Distill-Qwen-7B (Vanilla)**                            **[Total Tokens: 289]**

**<think> First, I need to determine the number of adults in the church. Since 40% of the 120 members are adults, I'll calculate $0.40 \times 120$, which gives me 48 adults. Next, I'll find the number of children by subtracting the number of adults from the total membership. So, $120 - 48$ equals 72 children. Finally, to find out how many more children there are than adults, I'll subtract the number of adults from the number of children: $72 - 48$, which equals 24. </think>**
**Solution: Let's determine how many more children there are than adults in the church. Total Members: The church has a total of 120 members. Number of Adults: 40% of the members are adults.**

$$\text{Number of Adults} = 0.40 \times 120 = 48$$

**Number of Children: The rest of the members are children.**

$$\text{Number of Children} = 120 - 48 = 72$$

**Difference:**

$$\text{Difference} = 72 - 48 = 24$$

**Final Answer:**

$$\boxed{24}$$

---

**DRQA-7B (Ours)**                                               **[Total Tokens: 82]**

**<think> 1. Adults: $120 \times 0.4 = 48$. 2. Children: $120 - 48 = 72$. 3. Difference: $72 - 48 = 24$. </think> The number of more children than adults is:**

$$\boxed{24}$$

## E.2 CASE STUDY 2: MATH-500

**Question**

**How many positive whole-number divisors does 196 have?**

---

**DeepSeek-R1-Distill-Qwen-7B (Vanilla)**        **[Total Tokens: 740]**

`<think>` **Alright, so I need to find how many positive whole-number divisors 196 has. Hmm, okay, I remember that to find the number of divisors, I need to factorize the number into its prime factors first. Let me see, 196 is a number I see a bit often, is it a square? Let me check. 14 squared is 196 because 14 times 14 is 196. Yeah, so that might be useful. Wait, instead of just stopping there, maybe I should confirm. If 14 times 14 is 196, then 14 is a factor of 196. But since 14 isn't a prime number, I guess I need to break it down further into prime factors. So, breaking down 14, it's 2 times 7, right? So, both 2 and 7 are primes. So, if 196 is 14 squared, then it's 14 times 14, which is (2 times 7) times (2 times 7). So, putting that together, 2 multiplied by 2 is $2^2$, and 7 multiplied by 7 is $7^2$. So, 196 is equal to $2^2 \times 7^2$. Okay, so now in prime factorization form, 196 is written as $2^2 \times 7^2$. I remember that the number of positive divisors is found by taking the exponents of each prime factor, adding one to each, and then multiplying them together. So, the formula is, if a number $N$ can be expressed as a product of primes where $N = p_1^a \times p_2^b \times \cdots \times p_n^k$, then the number of positive divisors is $(a+1)(b+1)\ldots(k+1)$. ... (omit) ...** `</think>`
**To find the number of positive whole-number divisors of 196, we start by performing the prime factorization of 196. First, we note that 196 is a perfect square, specifically $14^2$. Breaking down 14 into its prime factors, we get $2 \times 7$. Therefore:**

$$14^2 = (2 \times 7)^2 = 2^2 \times 7^2$$

**Using the formula for the number of divisors, if a number $N$ can be expressed as $p_1^a \times p_2^b \times \ldots \times p_n^k$, the number of positive divisors is $(a+1)(b+1)\ldots(k+1)$. For 196, the prime factorization is $2^2 \times 7^2$, so the exponents are 2 and 2. Applying the formula, we get:**

$$(2+1)(2+1) = 3 \times 3 = 9$$

**To verify, we list all the divisors of 196: 1, 2, 4, 7, 14, 28, 49, 98, and 196. Counting these, we find there are 9 divisors. Thus, the number of positive whole-number divisors of 196 is $\boxed{9}$.**

---

**DRQA-7B (Ours)**        **[Total Tokens: 165]**

**`<think>` We are asked: "How many positive whole-number divisors does 196 have?" Number: 196. Factorization: $196 = 14^2 = (2 \times 7)^2 = 2^2 \times 7^2$. Divisors count formula: $(a+1)(b+1)$. Calculation: $(2+1)(2+1) = 3 \times 3 = 9$. Answer: 9. `</think>`**
**Step 1: Prime Factorization**
$$196 = 2^2 \times 7^2$$

**Step 2: Count the Divisors The number of positive divisors is given by the formula:**

$$(2+1) \times (2+1) = 3 \times 3 = 9$$

**Final Answer**
$$\boxed{9}$$

