# OpenReview forum: "DRQA: Dynamic Reasoning Quota Allocation for Controlling Overthinking in Reasoning Large Language Models"
_ICLR.cc/2026/Conference — ICLR 2026 Conference Withdrawn Submission_

### Official Review · Reviewer_6vc4 · 2025-11-01

**Soundness:** 2
**Presentation:** 2
**Contribution:** 1
**Rating:** 2
**Confidence:** 4

**Summary:**

This paper tackles LLM overthinking by transferring the efficiency of batch inference into single-question settings. The authors discovered that batch inference settings encourage more concise reasoning, and scaling up the batch size improves reasoning efficiency. However, performing SFT on generations from batch inference mode leads to lower-quality model. Therefore, the authors propose DRQA, a method for encouraging concise reasoning while maintaining model quality. The authors build a preference dataset from a teacher model in batch vs. single inference mode and train the student model to predict whether the answer is correct and concise using GRPO. Across GSM8K, MATH-500, AIME, and LiveCodeBench, DRQA cuts token usage by 30% while maintaining or improving accuracy, outperforming a simpler SFT imitation approach that shortens outputs but can hurt accuracy on harder tasks.

**Strengths:**

1. The proposed method shows strong empirical effectiveness.
2. The method is well-motivated and clearly presented.

**Weaknesses:**

1. It's unclear how the proposed objective (classifying CoTs as accurate/concise) translates into better generation quality. Predicting labels may not directly improve the model's ability to produce accurate, succinct solutions.
2. The data source used for genearting concise CoTs is questionable. Batch inference is only one of many ways to elicit shorter CoTs and is not reliably shorter than single inference; using it as the primary mechanism to obtain concise traces seems brittle compared with alternative prompting or decoding strategies.
3. The labeling mechanism needs a redesign. Assigning label B to batch CoTs whenever the final answer is correct assumes those traces are “thorough and concise,” which does not necessarily follow; correctness alone is an insufficient proxy for reasoning quality.
4. The proposed approach is closer to an application of DPO-style, off-policy preference optimization than a novel RL innovation. It is essentially teaching the model to prefer accurate and more concise responses.
5. This paper may be offering limited novelty and insight. The contributions appear incremental, applying existing RL/preference-learning techniques with modest conceptual advances; the paper offers few new insights into when or why the method should outperform simpler baselines.
6. The method requires the reasoning chains of a larger model, whereas the baselines do not necessarily require them.

**Questions:**

Please see weaknesses.

---

> ### Author Response · Authors · 2025-11-30
>
> Dear Reviewer 6vc4,
>
> We appreciate your review and your acknowledgment of DRQA's empirical effectiveness. However, we believe there are significant misunderstandings regarding our training objective and the fundamental motivation of our method. We hope to clarify these points using the framework of "internalizing passive pressure into an active skill."
>
> **W1: Misunderstanding of the Training Objective (Classification vs. Generation)**
>
> The reviewer asks how "classifying CoTs" improves generation quality. This suggests a deviation in understanding our RL framework.
> *   **Correction:** We are **not** training a classifier to predict labels. We use GRPO to optimize the **generation policy**. The A/B/C labels serve solely as **reward signals**, not as output content.
> *   **The Problem:** Current RLLMs act like "nerds"—they overthink even simple queries (e.g., "5+8=?").
> *   **Our Solution:** DRQA uses these reward signals to teach the model to "internalize" the efficiency observed in batch processing. The model learns *which* reasoning patterns trigger the "concise and correct" reward, thereby actively improving the conciseness of its generation during inference.
>
> **W2: Validity of Batch Inference as a Data Source**
>
> The reviewer questions why we chose batching over other compression methods. In **Table 5**, we compared DRQA against a baseline labeled "**Qwen2.5-7B Data + RL**," which uses concise CoTs generated by a standard model without batching pressure.
> *   **Result:** DRQA significantly outperforms this baseline (** 79.58% vs. 76.76%**).
> *   **Reasoning:** We argue that batching introduces a physical **"attention budget" constraint**. Under this pressure, only high-saliency logic is preserved. While standard prompting often leads to random deletions, batching forces the model to filter for core logic, making it a superior source of high-quality training data.
>
> **W3: Effectiveness of the Labeling Mechanism (Is "Batch+Correct" really high quality?)**
>
> The reviewer questions assigning Label B to all correct batch CoTs. We argue that **correctness under pressure is the strongest indicator of efficiency.**
> *   **Survival of the Fittest:** On hard tasks (like AIME), raw batching often leads to over-compression, causing accuracy to drop by ~17%. Therefore, any reasoning chain that *survives* the harsh competition of batching and remains *correct* is, by definition, efficient and robust. "Low-quality" compressed chains are naturally filtered out because they become "Wrong" answers (Label C).
>
> **W4 & W5: Novelty and Insight**
>
> The reviewer suggests this is merely DPO-style optimization. We respectfully disagree. Our contribution is not the algorithm (GRPO is just a tool) but the **risk-aware preference framework** that solves the "blind compression" problem.
> *   **SFT (Imitation Learning) Failure:** As shown in Table 2, SFT leads to catastrophic forgetting on hard tasks because the model blindly mimics brevity without understanding the underlying logic.
> *   **DRQA Innovation:** Our A/B/C reward design teaches the model to weigh risks:
>     *   *Simple Questions:* The risk of error (Label C) is low, so the model pursues the high reward of Label B (Concise).
>     *   *Hard Questions:* The risk of error (Label C) is high. Since the penalty for "Wrong" outweighs the cost of "Verbose" (Label A), the model autonomously chooses the safer, detailed path.
> This is not just "preferring shorter answers"; it is a learned strategy for **Dynamic Reasoning Quota Allocation** based on difficulty.
>
> **W6: Dependence on Larger Models**
>
> The reviewer notes our use of DeepSeek-R1's reasoning chains as a limitation. We clarify this from two angles:
> 1.  **Self-Evolution:** DRQA does not strictly require a larger model. The core mechanism is that batch inference is more concise than single inference. Thus, DRQA can work entirely in a **self-distillation** setting (where Student = Teacher). We used DeepSeek-R1 in our experiments simply to leverage the best available open-source engine for SOTA results.
> 2.  **Comparison with Baselines:** Most SOTA efficient reasoning methods are essentially distillation or involve strong supervision. For instance, the DeepSeek-R1-Distill series is itself distilled from R1. In contrast, DRQA's data generation is extremely simple—requiring only batch inference runs—making it a highly cost-effective and scalable method for self-improvement.
>
> We hope this explanation clarifies that DRQA is a method for **internalizing efficient reasoning through risk management**, rather than a simple classification or imitation task.

---

### Official Review · Reviewer_bYMF · 2025-11-01

**Soundness:** 3
**Presentation:** 2
**Contribution:** 2
**Rating:** 4
**Confidence:** 4

**Summary:**

This paper proposes Dynamic Reasoning Quota Allocation (DRQA) to address the "overthinking" problem in reasoning models. The authors observed an interesting behavior: when RLLMs process multiple questions in batch mode, they are more resource-efficient and "dynamically compress reasoning steps for easier problems." Based on this observation, the authors built a reinforcement learning training dataset by generating both verbose and concise answers. Experiments show that when trained on the proposed dataset, the model can improve efficiency and accuracy and has better generalization.

**Strengths:**

- This paper targets "overthinking," a significant and practical issue in RLLMs where models are computationally inefficient.
- The observation of "resource competition pressure" is an interesting and novel motivation for the work.
- The method consistently achieves a "most favorable trade-off": while other methods can produce even shorter outputs, they often "suffer from severe accuracy degradation," a problem DRQA avoids.

**Weaknesses:**

- The paper's core motivation is the "resource competition pressure" observed in batch inference, but this mechanism is not directly applied to the method. The authors' method just aims to mimic the results of this finding, not the process itself.
- There is a lack of qualitative analysis of the "overthinking" phenomenon and how batch processing solves this by, for example, removing specific types of redundancy.
- The methodological contribution is somewhat thin; the paper argues, but does not definitively prove, that its specific preference data scheme is the best approach.
- The data source is only from DeepSeek-R1, so it needs more proof that the method works on other compression styles, not just on DeepSeek's.

**Questions:**

- How does your DRQA (Batch-2) model learn to selectively apply conciseness to simple problems while avoiding the failure mode on complex problems (like AIME) seen in your Batch-5 ablation?
- Have you considered using a dynamic token budget or a computational cost as a penalty in the RL objective, rather than using static, pre-generated batch outputs as preference data? This might more directly simulate the "resource competition" mechanism.
- Could you provide qualitative examples comparing a "Vanilla" response, a "Batch-2" response, and a final "DRQA" response for the same question? Specifically, what types of reasoning steps (e.g., arithmetic checks, definitions, verbal fluff) are being removed?

---

> ### Author Response · Authors · 2025-11-30
>
> Dear Reviewer bYMF,
>
> We sincerely appreciate your insightful feedback. We are particularly grateful for your affirmation of our "resource competition" motivation and for recognizing that DRQA achieves a favorable trade-off between efficiency and accuracy. Below are our detailed responses to your specific points:
>
> **W1 & Q2: Connection between "Resource Competition" and the RL Approach**
>
> The reviewer raises an excellent question: why not incorporate a dynamic token cost directly into the RL objective? We have, in fact, explored this through the *GRPO+Length Penalty* baseline (Table 3).
>
> *   **The Problem:** A direct penalty acts as a continuous signal that is difficult to tune. As shown in Table 3, this approach leads to over-compression on complex tasks and a drop in AIME accuracy, as the model indiscriminately learns that "shorter is better."
> *   **Our Solution (Internalization Mechanism):** Instead of imposing a coarse penalty on tokens during training, DRQA distills the complex outcomes of resource competition into a learnable policy using preference data. We transform the external physical pressure (context window constraints) into an internal value judgment—specifically, the trade-off between Options A and B. This enables the model to learn patterns of efficient reasoning in a manner that is significantly more stable and effective than simple length penalization.
>
> **W2 & Q3: Qualitative Analysis of Overthinking**
>
> We agree that qualitative examples demonstrate the value of our method. We have added a comparison between DRQA and original model responses in the Appendix. Our analysis shows that DRQA primarily eliminates two types of redundancy:
> 1.  **Self-doubt and Hesitation:** The original model often generates filler content like "Let me think... I need to be careful..." DRQA skips this and begins logical reasoning directly.
> 2.  **Redundant Verification of Simple Facts:** For steps like "2+3=5," the original model might output, "Let me verify: 2 plus 3 equals 5, yes that is correct." DRQA compresses this into direct execution: "2+3=5."
>
> Crucially, DRQA fully preserves the core logical steps required for solving complex problems.
>
> **W3: Methodological Contribution and Preference Scheme**
>
> We believe the A/B/C preference scheme is a significant contribution because it defines efficiency based on *semantics* rather than a simple metric based on *length*.
> *   **Empirical Support:** In Table 5, we compare DRQA against "Qwen2.5-7B Data + RL" (standard concise data). DRQA significantly outperforms this baseline, demonstrating that our specific preference scheme, derived from batch pressure, yields better results than generic "short" data.
>
> **W4: Data Source Bias (DeepSeek-R1)**
>
> We selected DeepSeek-R1 as it is currently the state-of-the-art open-source teacher model. To demonstrate generalization, we successfully applied DRQA to **DeepSeek-R1-Distill-Llama-8B** (Table 3). We observed consistent performance improvements across both Qwen and Llama architectures. This indicates that our method addresses the general "overthinking" behavior found in RLLMs, rather than being tailored to a specific pattern in DeepSeek models.
>
> **Q1: How DRQA Avoids Failure on Complex Problems (e.g., vs. Batch-5)**
>
> This highlights the core advantage of our RL framework.
> *   **Failure of Batch-5 (SFT):** SFT forces the model to mimic the concise outputs of Batch-5. On hard problems, Batch-5 outputs are often *too short* and *incorrect* (as we noted to other reviewers, the original Batch-5 accuracy on AIME drops to ~57%). SFT blindly copies this "bad" behavior.
> *   **Success of DRQA (RL):** DRQA learns a risk management strategy through our reward structure:
>     *   Option B (Concise & Correct) -> High Reward.
>     *   Option C (Wrong) -> High Penalty.
>
> On complex problems (like AIME), the model implicitly predicts that attempting conciseness (Option B) carries a high risk of error (Option C). Consequently, it opts for the safer, more verbose path (Option A) or a balanced approach. Effectively, it learns: **"Be concise only when you are confident you won't be wrong."**
>
> We hope this response clarifies how DRQA internalizes the resource competition mechanism to achieve robust and adaptive reasoning capabilities.

---

### Official Review · Reviewer_Ua16 · 2025-11-01

**Soundness:** 2
**Presentation:** 3
**Contribution:** 2
**Rating:** 4
**Confidence:** 3

**Summary:**

The paper proposes Dynamic Reasoning Quota Allocation (DRQA) to reduce overthinking in reasoning large language models. It is inspired by the observation that batch inference naturally shortens reasoning chains through implicit resource competition. DRQA transfers this behavior to single-question inference using reinforcement learning with preference-labeled data. The model learns to prefer concise and accurate reasoning chains over verbose ones via Group Relative Policy Optimization (GRPO). Across multiple reasoning benchmarks, DRQA cuts token usage by 25–35% while maintaining or improving accuracy. It outperforms baselines like DAST and GRPO+Length Penalty on both mathematical and code tasks. Overall, the paper presents a simple yet effective framework for adaptive and efficient reasoning in large models.

**Strengths:**

- The paper identifies and formalizes a previously underexplored phenomenon in reasoning LLMs that batch inference implicitly encourages concise reasoning through token competition.
- The proposed Dynamic Reasoning Quota Allocation (DRQA) effectively bridges the batch and single-question reasoning paradigms using reinforcement learning with preference data.
- DRQA is evaluated across diverse and comprehensive reasoning benchmarks, multiple model sizes, and ablation settings.

**Weaknesses:**

- While empirically demonstrated, the notion of “resource competition pressure” lacks a clear theoretical or mechanistic explanation. The current argument remains descriptive, leaving ambiguity about whether the effect stems from model inductive bias, context compression, or decoding heuristics.

- Reward formulation is indirect and loosely tied to reasoning efficiency. DRQA trains a classifier to label reasoning chains as A/B/C (correct-concise vs. correct-verbose vs. incorrect) and uses GRPO to favor the correct label. However, the reward signal is categorical and not proportional to actual reasoning length or quality. Thus, it cannot finely distinguish between moderately concise and overly compressed reasoning, potentially leading to unstable learning or suboptimal trade-offs.

- Limited justification for using GRPO instead of more targeted RL objectives. Why did this paper choose GRPO instead of other RL losses?

- DRQA’s claim of “adaptive allocation” would be stronger if the authors stratified test questions by difficulty (e.g., easy vs. hard GSM8K) and showed differing average reasoning lengths. Without this, the adaptive claim remains qualitative.

- All training data stem from DeepSeek-R1 outputs, which might bake in its reasoning biases. The paper does not explore whether DRQA generalizes when the teacher is of a different architecture (e.g., Qwen2.5).

**Questions:**

Please refer to the weakness.

---

> ### Author Response · Authors · 2025-11-30
>
> Dear Reviewer Ua16,
>
> We sincerely appreciate your recognition of the novelty regarding the "resource competition" phenomenon and the effectiveness of the DRQA framework. We are grateful for your constructive feedback regarding the underlying mechanism and adaptive capabilities. Below are our point-by-point responses:
>
> **W1 (Theoretical Mechanism of "Resource Competition Pressure")**
>
> We acknowledge that our initial explanation was primarily empirical. We hypothesize that this phenomenon stems from an **"attention budget" mechanism under context constraints**. When processing batched queries, the shared context window acts as a bottleneck; to maintain coherence across multiple logical streams, the model's attention heads are forced to prioritize high-saliency tokens (core logic) while suppressing low-information tokens (redundant verbiage). While a full interpretability study is reserved for future work, we have added a discussion of this theoretical hypothesis in Section 2 to strengthen the theoretical grounding of the observed phenomenon.
>
> **W2: Rationality of Categorical Rewards (A/B/C) vs. Continuous Length Penalties**
>
> We deliberately chose a semantic-based categorical reward over a length-based continuous reward for the sake of robustness. Continuous penalties (e.g., R - alpha*L) are highly sensitive to hyperparameters. As shown by the *GRPO+Length Penalty* results in Table 3, penalty strengths suitable for simple tasks often lead to over-compression on complex tasks, forcing the model to prune necessary reasoning steps. Our A/B/C classification mechanism allows the model to learn the qualitative concept of "concise *and* accurate." It encourages the model to pursue conciseness only when it is safe to do so, thereby avoiding the risk of being "Wrong" (Category C).
>
> **W3: Justification for Using GRPO**
>
> We selected Group Relative Policy Optimization (GRPO) primarily for two key reasons:
> 1.  **Efficiency:** It eliminates the need for a separate value function critic, significantly reducing memory overhead.
> 2.  **Relative Baseline Mechanism:** The group-wise comparison aligns naturally with our preference learning setup and offers a more stable training process compared to standard PPO.
>
> **W4: Evidence of "Adaptive Allocation" (Stratification by Difficulty)**
>
> Thank you for this valuable suggestion. We demonstrate the adaptive nature of our model by analyzing the absolute token allocation across tasks of varying difficulty in our existing results (Table 3):
> *   **Difficulty Stratification:** We use GSM8K to represent easy tasks and AIME to represent hard tasks.
> *   **Adaptive Behavior:**
>     *   *Easy Task (GSM8K):* DRQA allocates approximately **1,400 tokens** on average.
>     *   *Hard Task (AIME 2024):* DRQA allocates approximately **11,000 tokens** on average.
> *   Comparing Table 2 (SFT) and Table 3 (DRQA) further confirms this. The SFT approach, which mimics raw batching, suffers a significant accuracy drop on AIME (74.67% -> 57.33%) due to blind compression. In contrast, DRQA maintains high accuracy on AIME while actively cutting redundancy on simple tasks, preserving necessary depth for complex ones.
> *   **Conclusion:** Model correctly recognizes that AIME presents significantly more challenging problems and maintains substantially longer reasoning chains accordingly.
>
> **W5: Generalization and Teacher Model Bias**
>
> We utilized DeepSeek-R1, currently the state-of-the-art open-source reasoning model. To ensure our method is not limited to a specific architecture, in addition to Qwen, we successfully applied DRQA to **DeepSeek-R1-Distill-Llama-8B**. As shown in Table 3, DRQA achieves consistent improvements across both architectures, demonstrating the robustness of our approach.
>
> We hope this response clarifies the rationale behind our reward design and sufficiently demonstrates the model's adaptive nature through our experimental results.

---

### Official Review · Reviewer_XmWr · 2025-11-01

**Soundness:** 3
**Presentation:** 3
**Contribution:** 2
**Rating:** 4
**Confidence:** 4

**Summary:**

This paper proposes DRQA, a method that leverages the resource competition phenomenon when asking multiple questions together to generate training data, which is then trained with GRPO to make LRMs more efficient.

This reviewer happened to review a prior version of this work at another conference, which the authors have then withdrawn without rebuttal. Many portion of my review are copied with notes on how the authors address them in the updated version.

**Strengths:**

* Good writing.
* Good baseline coverage on efficient reasoning methods.
* Good coverage on reasoning tasks.
* The idea of adopting prompt batching to compress reasoning traces seems interesting.

**Weaknesses:**

* Lack of novelty and insufficient connection to existing literature. The idea of putting multiple questions into the same prompt has already been well explored in existing literature [1, 2, 3] and likely more, but none are cited or discussed. Since DRQA is largely a combination of this prompt batching idea and GRPO, the paper oversells its novelty. Its relation to existing methods deserves much better coverage.
    * While the updated version does cite some of these works, they are still not properly discussed in the Related Work section but only (mainly?) briefly mentioned in the introduction. Given the close connection between the proposed work and these existing studies, this treatment feels overly hand-wavy and borderline disrespectful—especially considering that the authors added these citations after reading this reviewer’s previous feedback. This suggests the lack of discussion was not due to oversight, but rather a deliberate choice.


* Limited experiment coverage on prompt compression data. DRQA is motivated by the "resource competition" phenomenon, but the end goal still seems to be producing answers of varying conciseness for the same question, as resource for the later fine-tuning. In this setting, a broader comparison against established prompt compression techniques is necessary, but almost none are included.
   * Seems to remain unaddressed as no prompt compression-based training baseline is provided.

* Insufficient model coverage. Only two small Deepseek-R1-Distilled models are evaluated. Even within this narrow scope, the gains over the 2 GRPO baselines at 7B are relatively marginal.
    * Seems only a DeepSeek-R1-Distilled-Llama-7B is added, so still limited.

* Narrow dataset coverage outside the math domain. Since the model is trained on large amounts of DeepScaleR data (which includes problem–answer pairs from AIME, AMC, etc.), most evaluation remains heavily in-domain. Broader coverage on out-of-domain tasks—such as coding, planning, and others—is needed.
    * Mostly resolved with the inclusion of LiveCodeBench and GPQA-Diamond. One might nitpick that LiveCodeBench results do not feature many baselines.

Again, this reviewer generally like this method, if the author can show that generating training data with prompt batching can lead to higher quality (for instance, better than training on compressed CoTs) for most LRMs (beyond DS distill and on bigger models), this reviewer is open to bumping the scores pending on proper discussion of related work and other presentation fixes.

[1] Batch Prompting: Efficient Inference with Large Language Model APIs
[2] BatchPrompt: Accomplish More with Less
[3] Auto-Demo Prompting: Leveraging Generated Outputs as Demonstrations for Enhanced Batch Prompting

**Questions:**

Same as my previous questions/comments:

* How are baselines like AdaptThink and AutoL2S done? These methods are training non-reasoning models to be efficient LRMs, where DRQA is compressing LRMs.
* The real baseline under this paper is not Vanilla, but GRPO, as the DRQA is trained with additional data. This should be highlighted and featured in places like F4.

---

> ### Author Response · Authors · 2025-11-30
>
> Dear Reviewer XmWr,
>
> We appreciate your constructive feedback and your correction regarding the relevance of prior batch prompting research. We value this opportunity to clarify our research contributions and implementation details. Below are our point-by-point responses to your specific concerns:
>
> **W1 (Lack of novelty):**
>
> DRQA is not merely a simple combination of "prompt batching" and GRPO. While we leverage the batching phenomenon, our core contribution lies in transforming this passive, context-dependent pressure into an active, internalized skill.
> Passive vs. Active: As shown in Table 2, attempting to make the model mimic the compression behavior of batch prompting via SFT leads to catastrophic forgetting (with AIME accuracy dropping by approximately 17%). This occurs because the model blindly mimics brevity without understanding when to be concise.
> Risk Management Mechanism: Our novel preference construction ("Correct but Verbose" vs. "Correct and Concise" vs. "Wrong") teaches the model dynamic risk assessment. For simple queries, the model seeks the "conciseness" reward (low risk); for complex queries, it prioritizes "correctness" (high error risk) over brevity. This adaptive mechanism, derived from our specific RL design, is fundamentally different from system-level batching optimizations [1, 2, 3]. DRQA successfully enables the model to internalize this adaptive allocation ability, allowing it to dynamically manage its "thinking" quota in single-question inference, much like an expert.
>
> **W1 (Regarding the discussion of existing batch prompting literature):**
>
> We apologize if the placement of citations [1, 2, 3] appeared dismissive; this was not our intention. In the revised version, we will move these citations to the "Related Work" section and explicitly articulate the distinction: existing research focuses on system-level efficiency (maintaining throughput and latency), whereas DRQA focuses on model-level reasoning density (mitigating overthinking).
>
> **W2: (Comparison with training on prompt compression data)**
>
> Regarding the comparison with "training on prompt compression data," please refer to the ablation study in Table 5. We compared DRQA against a baseline labeled "Qwen2.5-7B Data + RL," which uses concise reasoning generated directly by Qwen2.5-7B.
> Results: DRQA significantly outperforms this baseline (Overall Accuracy 79.58% vs. 76.76%). This empirically proves that "resource competition" during batching produces high-quality, logic-preserving compression that standard concise prompting cannot replicate.
>
> **W3 & W4 (Model and Dataset Coverage):**
>
> We utilized the DeepSeek-R1-Distill series (Qwen-1.5B/Qwen-7B/Llama-8B). We selected models of two different sizes (1.5B and 7/8B) and two different architectures (Qwen and Llama) to demonstrate the generalizability of our method. Regarding larger models, limited by computational resources during the rebuttal period, we plan to extend the evaluation to larger-scale models in the final version.
>
> We have included LiveCodeBench and GPQA-Diamond to demonstrate DRQA strong out-of-distribution generalization beyond mathematical tasks. Additionally, addressing the reviewer's comment regarding the need for more comparative results on LiveCodeBench, we have added results for GRPO and GRPO+Length Penalty in the revised version.
>
> **Q1: Implementation of baselines like AdaptThink and AutoL2S**
>
> We strictly evaluated AutoL2S and AdaptThink using their officially released model checkpoints to ensure faithful reproduction.
> We agree with the reviewer’s point that these methods are fundamentally different. AutoL2S trains base models (e.g., Qwen2.5-7B) to achieve efficient reasoning, whereas DRQA and AdaptThink compress stronger RLLMs (e.g., DeepSeek-R1-Distill). We retained these baselines to map a comprehensive landscape of "efficient reasoning" techniques. In the revised version, we explicitly categorize the baselines and highlight the differences in initial checkpoints to ensure a fair comparison within the appropriate context.
>
> **Q2: The real baseline should be GRPO rather than the Vanilla model**
>
> We fully agree. This is exactly why we feature GRPO (standard reward) and GRPO+Length Penalty as the primary baselines in Table 3.
> Comparison Results: Taking the 1.5B model as an example, DRQA achieves an overall accuracy of 55.69%, outperforming standard GRPO (55.66%) while significantly reducing token usage (6801 vs. 9787).
> Conclusion: DRQA outperforms standard GRPO because standard GRPO lacks a "conciseness" incentive, whereas GRPO+Penalty harms accuracy by overly penalizing complex reasoning (dropping to 51.93%). DRQA strikes the optimal balance.
>
> We hope this response clears up your concerns regarding our baseline implementation and the novelty of DRQA.

---

### Note · Authors · 2026-01-11

I have read and agree with the venue's withdrawal policy on behalf of myself and my co-authors.